# Immobilization and Application of the Recombinant Xylanase GH10 of *Malbranchea pulchella* in the Production of Xylooligosaccharides from Hydrothermal Liquor of the Eucalyptus (*Eucalyptus grandis*) Wood Chips

**DOI:** 10.3390/ijms232113329

**Published:** 2022-11-01

**Authors:** Robson C. Alnoch, Gabriela S. Alves, Jose C. S. Salgado, Diandra de Andrades, Emanuelle N. de Freitas, Karoline M. V. Nogueira, Ana C. Vici, Douglas P. Oliveira, Valdemiro P. Carvalho-Jr, Roberto N. Silva, Marcos S. Buckeridge, Michele Michelin, José A. Teixeira, Maria de Lourdes T. M. Polizeli

**Affiliations:** 1Departamento de Biologia, Faculdade de Filosofia, Ciências e Letras de Ribeirão Preto, Universidade de São Paulo, Ribeirão Preto 14040-901, SP, Brazil; 2Departamento de Bioquímica e Imunologia, Faculdade de Medicina de Ribeirão Preto, Universidade de São Paulo, Ribeirão Preto 14049-900, SP, Brazil; 3Departamento de Química, Faculdade de Filosofia, Ciências e Letras de Ribeirão Preto, Universidade de São Paulo, Ribeirão Preto 14040-901, SP, Brazil; 4Departamento de Química e Bioquímica, Faculdade de Ciências e Tecnologia, Universidade Estadual Júlio de Mesquita Filho, Presidente Prudente 19060-900, SP, Brazil; 5Laboratório de Fisiologia Ecológica de Plantas, Departamento de Botânica, Instituto de Biociências, Universidade de São Paulo, São Paulo 05508-090, SP, Brazil; 6CEB—Centre of Biological Engineering, University of Minho, Campus Gualtar, 4710-057 Braga, Portugal; 7LABBELS—Associate Laboratory, University of Minho, Campus Gualtar, 4710-057 Braga, Portugal

**Keywords:** immobilization, xylanases, xylooligosaccharides, *Malbranchea pulchella*, *Eucalyptus grandis*, hydrothermal liquor

## Abstract

Xylooligosaccharides (XOS) are widely used in the food industry as prebiotic components. XOS with high purity are required for practical prebiotic function and other biological benefits, such as antioxidant and inflammatory properties. In this work, we immobilized the recombinant endo-1,4-β-xylanase of *Malbranchea pulchella* (MpXyn10) in various chemical supports and evaluated its potential to produce xylooligosaccharides (XOS) from hydrothermal liquor of eucalyptus wood chips. Values >90% of immobilization yields were achieved from amino-activated supports for 120 min. The highest recovery values were found on Purolite (142%) and MANAE-MpXyn10 (137%) derivatives, which maintained more than 90% residual activity for 24 h at 70 °C, while the free-MpXyn10 maintained only 11%. In addition, active MpXyn10 derivatives were stable in the range of pH 4.0–6.0 and the presence of the furfural and HMF compounds. MpXyn10 derivatives were tested to produce XOS from xylan of various sources. Maximum values were observed for *birchwood* xylan at 8.6 mg mL^−1^ and wheat arabinoxylan at 8.9 mg mL^−1^, using Purolite-MpXyn10. Its derivative was also successfully applied in the hydrolysis of soluble xylan present in hydrothermal liquor, with 0.9 mg mL^−1^ of XOS after 3 h at 50 °C. This derivative maintained more than 80% XOS yield after six cycles of the assay. The results obtained provide a basis for the application of immobilized MpXyn10 to produce XOS with high purity and other high-value-added products in the lignocellulosic biorefinery field.

## 1. Introduction

In recent years, the biomass biorefinery concept has been extended beyond the technological application of bioethanol production. Innovative and efficient technologies for lignin, cellulose, and hemicellulose fractionation allow the implementation of integrated processes for the co-production of bioenergy and higher value-added bioproducts, such as cello- and xylooligosaccharides and lignin derivatives [1,2,3,4]. These strategies are necessary to maximize organic matter, reduce waste generation, and obtain products with high added-value, strengthening the circular bioeconomy in plant biomass biorefineries [5].

Xylan is the most common hemicellulosic polysaccharide in the plant cell wall and the second most abundant polysaccharide in nature [6]. It is constituted mainly of xylose monomers joined with β-1,4 glycosidic bonds. It can be found as complex polysaccharides comprising a homopolymeric backbone of β-1,4-linked D-xylopyranosyl units substituted with D-glucuronopyranosyl, 4-O-methyl-D-glucuronopyranosyl, L-arabinofuranosyl, acetyl, feruloyl, and *p*-coumaroyl side-chain groups [6,7]. The complete hydrolysis of the complex heteropolymer of xylan requires the activity of several enzymes, which act cooperatively to convert the heteropolymer into monomeric units. Endo-1,4-β-xylanases (EC 3.2.1.8) are the main enzymes of the xylanolytic system, cleaving β-1–4 glycosidic bonds from xylan and releasing xylooligosaccharides (XOS) of varied sizes. The β-xylosidases (EC 3.2.1.37) act with the endo-1,4-β-xylanases, releasing xylose units from the non-reducing ends of the XOS. Furthermore, the hydrolysis of certain xylans may require the action of other enzymes like α-arabinofuranosidases (EC 3.2.1.55), α-glucuronidases (EC 3.2.1.139), acetyl xylan esterases (EC 3.1.1.72), feruloyl esterases (EC 3.1.1.73), and *p*-coumaric acid esterases (EC 3.1.1.-) [2,7].

Among the enzymes of the xylanolytic system, the interest in xylanases has grown remarkably due to their application in the hydrolysis of lignocellulosic materials (LCM), releasing XOS (sugar oligomers composed of xylose units linked with β-(1→4) bonds) [7,8,9]. XOS have been preferentially used as prebiotic components in developing new functional foods, acting as a regulatory substrate for intestinal flora, mainly XOS with a xylose degree of polymerization (DP) of 2–4 [9,10,11,12,13,14]. Furthermore, antioxidant, inflammatory, anticancer, and immunomodulatory activities have been reported as additional biological benefits of XOS [8,10]. Nowadays, XOS are mainly produced through chemical hydrolysis of xylan (acid hydrolysis or alkali extraction). However, alternative methods have been employed, such as autohydrolysis and enzymatic hydrolysis [7,9,10]. The enzymatic method is described as a promising strategy due to the high specificity of the enzymes to different types of xylans, which avoid or minimize the formation of undesirable products in the XOS mixture, such as xylose and furfural [7,9,10].

Recent works have demonstrated the use of free and immobilized xylanases in the hydrolysis of xylan, mainly from agricultural crop residues [8,10,13,14,15,16,17,18,19,20,21]. Thus, the search for better xylanases and the improvement of these biocatalysts to produce XOS through enzymatic hydrolysis has increased in the last few years [7,9,10]. However, the enzymatic hydrolysis of the xylan present in the LCM is still a great challenge, requiring studies on the composition and pretreatment of LCM besides efficient biocatalysts. Furthermore, obtaining and using these biocatalysts must be cost-effective to be economically viable [5,22]. It is well-established that the immobilization of enzymes is fundamental for the viability of the bioprocess since this technique allows the recovery and reuse of the enzyme through several reaction cycles [23,24,25,26]. Furthermore, the immobilization process may improve the activity, stability, and selectivity of the enzyme, desirable features for utilization in LCM biorefinery [24,27].

Here, the thermostable MpXyn10 xylanase from *Malbranchea pulchella* [15] was overexpressed in *Aspergillus nidulans* and immobilized in different supports activated with different functional groups. The activity, thermostability, and operational stability of the MpXyn10 immobilized were studied, and the analysis was applied in the XOS production from xylan of diverse sources. Finally, as a proof of concept in the LCM biorefinery, the potential of MpXyn10 in producing XOS from hydrothermal liquor of eucalyptus wood chips (*Eucalyptus grandis*) was evaluated.

## 2. Results and Discussion

### 2.1. Expression and Purification of Recombinant MpXyn10 Xylanase

Heterologous expression of the MpXyn10 xylanase was carried out in batch fermentation using a stirred tank bioreactor of 5 L. *Aspergillus nidulans* A773 was cultivated in a minimal medium supplemented with 5% corn syrup for 60 h at 37 °C. The xylanase-rich crude extract was filtered and concentrated, and as expected, high levels of xylanase activity (>200 U mL^−1^) were obtained. Afterward, MpXyn10 was purified from concentrated crude extract using two chromatography steps, as described in Methods. SDS–PAGE analysis (Figure 1) of the soluble fraction confirmed a single band with a molecular weight of approximately 49 kDa corresponding to MpXyn10 xylanase [15]. The specific activity of MpXyn10 was 178 ± 6 U mg^−1^. This purified fraction was used in the immobilization experiments.

### 2.2. Preparation and Characterization of Supports

MpXyn10 was immobilized on commercial Purolite and agarose-activated supports. These approaches were chosen to evaluate the best support/protocol to preserve the enzymatic activity and improve operational features of the immobilized MpXyn10, such as pH and temperature stability. Initially, hydroxyl groups on agarose surface were activated with epichlorohydrin, forming epoxy/diols groups, and then activated with several functional groups and characterized using FTIR spectroscopy. As shown in Figure 2A, it was possible to observe bands corresponding to the stretching vibration of bound water (O-H) at 1641 cm^−1^, of an alkene methylene group (C-H) at 1370 cm^−1^, and of glycosidic linkage (C-O-C) at 1059 cm^−1^, even as bands corresponding to epoxy rings around 1252, 930, and 890 cm^−1^ were observed [28,29]. The band at 2864 cm^−1^ was attributed to the elongation mode (=C-H) of the aldehyde (glyoxyl) group. For agarose activated with amino groups, a new band was formed at 3077 cm^−1^ attributed to amine stretching (N-H) [30,31]. For GLUT-agarose, the typical band at 1612 cm^−1^ confirms the presence of the imine (C=N) in the support arising from the bond between glutaraldehyde and amino groups of ethylenediamine (EDA) (Figure 2A) [32,33,34]. Concerning commercial support Purolite, the FTIR spectrum is shown in Figure 2B. Characteristic bands confirmed the groups present in the support, such as methyl group (C-H) stretching at 2960 cm^−1^, aliphatic ketone (C=O) stretching at 1717 cm^−1^, alkene (C=C) stretching at 1640 cm^−1^, alkene methylene group (C-H) bending at 1455 cm^−1^, alkene methyl group (C-H) bending at 1388 cm^−1^, amine (C-N) stretching at 1255 cm^−1^, aliphatic ether (C-O) stretching at 1147 cm^−1^, and alkene (C=C) bending at 960 cm^−1^ [35].

### 2.3. Immobilization of the MpXyn10 Xylanase

Figure 3 shows the time course of immobilization of MpXyn10 on different supports. Higher values of immobilization yield (>90%) were obtained on amino-activated supports (Purolite, MANAE, and PEI-agarose) in 120 min (Figure 3A). After that, a negligible amount of MpXyn10 was immobilized on these supports. The activity recovery (AR) was >100% for all amino-activated supports, highlighting the Purolite-MpXyn10 derivative with an AR of 142% (Table 1). These results suggest that the ionic exchange immobilization presented a slight effect on the enzyme and increased its activity after the immobilization. Its features are commonly related to hyperactivation, which indicates that the enzyme was immobilized in an activated form or that it improved the catalytic properties with respect to the corresponding free form [27]. According to Rodrigues et al. [36], these alterations in enzyme properties may be associated with changes in the enzyme structure or are characteristics of the support. In the case of the immobilization of xylanases, hyperactivation was usually observed after the immobilization of the enzyme from physical adsorption on ion-exchange supports, such as MANAE and carboxymethyl (CM) [36,37,38,39]. In this work, the hyperactivation observed for Purolite-MpXyn10 and MANAE-MpXyn10 derivatives, which most likely occur through the adsorption of MpXyn10 on free amino groups (-NH_2_) present in the surface of the support [27,40]. Its rich surface allows the immobilization of the enzyme in a dispersed phase and decreases the adsorption of enzyme clusters or aggregates, which greatly decreases the activity of the derivative [36]. Similar results were reported in the immobilization of the β-galactosidase from *Aspergillus oryzae* and β-xylosidase from *Bacillus subtilis* [41,42].

The activity recovery was low for glyoxyl- and GLUT-agarose derivatives (AR of 51 and 61%, respectively). It suggests that the immobilization of MpXyn10 from covalent attachment (one- and multipoint) via aldehyde groups, even at mild conditions (pH 7.0 and low ionic strength), promoted a negative effect on the enzyme activity upon immobilization on these supports (Table 1). The negative effect may be associated with the rigidification of the enzyme through one region or by random immobilization, distorting the active tridimensional structure of the enzyme [43,44,45]. In the specific case of MpXyn10, the attachment may develop through the cellulose-binding domain (CBM) of the enzyme. Previous studies showed that CBM provides an anchor for the MpXyn10 on the natural substrates (xylan oat spelt and avicel), contributing directly to the xylanolytic activity [15].

Considering the retention of activity (Table 1), the MANAE- and Purolite-MpXyn10 derivatives were chosen for the remaining studies.

### 2.4. Characterization of the MpXyn10 Immobilized Derivatives

Thermal stability assays were performed at 65 and 70 °C. Thermal inactivation profiles of the free and MpXyn10 derivatives after incubation for 24 h are shown in Figure 4. MANAE-MpXyn10 retained 90% of its activity at 65 °C and more than 80% at 70 °C (Figure 4). Under the same incubation conditions, Purolite-MpXyn10 maintained 80% of its activity at 65 °C and around 60% at 70 °C, while the residual activity of free MpXyn10 was approximately 60% at 65 °C but only 11% at 70 °C (Figure 4B). The MpXyn10 stability at high temperatures is therefore enhanced upon immobilization, as shown for other xylanases [21,37,46]. These results indicated that MpXyn10 derivatives are robust and stable biocatalysts and can be applied in reactions at elevated temperatures, which may be necessary to hydrolyze high xylan concentrations and increase reaction yields [37].

We also assessed the effect of pH on the stability of the MpXyn10 derivatives with incubation in buffers ranging from pH 4.0 to pH 6.0 (Figure 4C), conditions used in the enzymatic hydrolysis of xylan and LCM [3,10]. MANAE- and Purolite-MpXyn10 derivatives showed a similar profile observed for free MpXyn10, with residual activities around 90% after 24 h of incubation at 25 °C, with maximum activity at pH 5.0 (Figure 4C). Regarding the effect of furfural and HMF on MpXyn10 activity, no adverse effect on the activity of any of the preparations was observed (Figure 4D). These results are interesting since furfural and HMF may inhibit the activity of cellulases and hemicellulases, resulting in lower reaction yields [47]. Thus, MpXyn10 derivatives could be applied in processes in which these compounds are formed due to the higher pretreatment severity (time, temperature) used. Therefore, the immobilization step improved the activity and stability of MpXyn10, necessary features for the application in the hydrolysis of LCM. Hence, the potential of MpXyn10 derivatives for XOS production from xylan of various sources was investigated.

### 2.5. Xylan Enzymatic Hydrolysis

The hydrolysis profile of free and immobilized MpXyn10 was initially evaluated using xylan *beechwood* as a substrate (Figure 5A,B). No difference was observed in the hydrolysis profile for the immobilized MpXyn10, with a similar amount of reducing sugars released after 180 min of reaction (Figure 5A). In addition, analysis of the hydrolysis products obtained at different reaction times by TLC showed spots corresponding to XOS (DP 2–6) and barely visible spots corresponding to xylose after 3 h of the assay (Figure 5B). Among the XOS produced, the main product obtained was xylobiose (7.8 mg mL^−1^) for free and immobilized MpXyn10 preparations assayed (Table 2). These results are according to previous work on the characterization of MpXyn10 [15] and suggest that the hydrolysis pattern is not altered when free MpXyn10 is immobilized in the derivates. The profile of the hydrolysis products observed was similar for all xylans assayed (Figure 5), with the highest products being xylobiose (X2) and xylotriose (X3), respectively (Table 2). On the other hand, the xylose release profile observed was distinct after 3 h of the assay, according to the xylan source (Figure 5B,D). Indeed, it is well described in the literature on different xylan compositions, mainly regarding the type of substituents [7]. For example, xylan from *beechwood* and *birchwood* present mainly glucuronic acid as substituents and a high degree of acetylation; whereas wheat arabinoxylan and xylan from oat spelt are rich in arabinoses as substituents of the xylose main chain [48,49,50]. Furthermore, the production of xylose also is attributed to conditions of the assay, such as time, temperature, and amount of substrate, which may induce the enzyme to act on the terminal glycosidic bonds, releasing xylose [9,51]. Therefore, the results obtained confirmed that MpXyn10 is a GH10 family member and is active on a wide range of xylan substrates, releasing mainly XOS with a low degree of polymerization (DP 2–4) [8,15].

Among the biocatalysts evaluated, an XOS yield of around 50% was obtained for all xylans assayed using Purolite-MpXyn10 (Table 2). Maximum values of XOS were found for *birchwood* xylan (8.6 mg mL^−1^) and wheat arabinoxylan (8.9 mg mL^−1^) after 180 min of the assay (Table 2). Although the comparison with other works is difficult due to differences in substrates and XOS-quantification methodology, our results of XOS production agree with other works that applied immobilized xylanases, mainly due to the higher production of X2-X4 [21,37,46,52,53]. Thus, MpXyn10 derivatives were efficient in the production of XOS from different xylans without affecting the catalytic properties of MpXyn10, which makes its use promising in an LCM biorefinery context.

### 2.6. Production of XOS from Hydrothermal Liquor

*Eucalyptus* spp. are fast-growing hardwood used mainly in the production of cellulose and paper, generating many rich-cellulose residues [54,55], which classifies it as a promising feedstock for bioethanol production and its derivates [55,56]. In the present work, *Eucalyptus grandis* wood chips (EWC) were used, and the chemical composition of raw material was performed (Table 3). As shown in Table 3, the main sugars-derived components found in raw EWC were cellulose (33.4%) and xylan (14.8%), representing approximately 50% of EWC’s total composition (Table 3). This result is according to previous chemical composition reported for *E. grandis* residues and other species of *Eucalyptus*, such as *Eucalyptus nitens* [54] and *Eucalyptus globulus* [55,56], which also described this feedstock as successful in the bioethanol production.

LCM pretreatment is crucial to disrupting the lignocellulosic matrix and increasing hemicellulose sugar recovery, allowing higher cellulose accessibility and yields of fermentable sugars [57]. Liquid hot water (LHW) has become very popular in biorefinery since it only uses water at elevated temperatures and pressure [47,55,57]. Thus, the use of LHW in EWC was proposed as the first step to solubilize hemicelluloses, aiming to utilize and valorize EWC. As a proof of concept, LHW was carried out at 160 °C for 60 min (severity factor = 3.54). These conditions were used to obtain a good recovery of the soluble xylan-rich liquor after the pretreatment and to avoid the formation of unwanted products, such as furfural and 5-(hydroxymethyl)furfural from sugar degradation and even xylose. As shown in Table 3, LHW treatment showed good efficiency in the partial solubilization of hemicellulose from the raw EWC under the mildest studied condition. Xylan content reduced from 14.8 to 9.9% in the EWC pretreated solids, recovering 2.3 g/L solubilized XOS in the hydrothermal liquor and a low formation of sugar degradation compounds (Table 3).

After LHW pretreatment, enzymatic hydrolysis of xylan (solubilized as oligomers) present in EWC hydrothermal liquor was used to obtain XOS with lower xylose DP (2–4), which are preferentially used as prebiotic components in functional foods. Initially, free and immobilized MpXyn10 preparations were applied in the hydrolysis of EWC hydrothermal liquor, and the profile of the enzymatic hydrolysis products was analyzed by TLC and quantified by HPLC. As observed in Figure 6A, free and immobilized MpXyn10 derivatives produced XOS with low DP from EWC liquor (Figure 6A), X2 being the main product found (Figure 6B), with the maximum XOS production found using Purolite-MpXyn10, with X2 (0.68 mg mL^−1^) as the highest product obtained (Figure 6B). These results show the potential of Purolite-MpXyn10 to be used in the production of low DP XOS from diverse xylan sources.

### 2.7. Operational Stability of Purolite-MpXyn10

As one of the main disadvantages of enzyme processes to produce XOS is the cost of enzymes, therefore, the use of immobilized xylanases to increase catalyst efficiency is attractive. Generally, this strategy allows the development of more stable biocatalysts that can be recovered and reused several times. Thus, in order to evaluate the operational stability of the Purolite-MpXyn10, this derivative was used in several cycles of hydrolysis of xylan from *beechwood* and EWC hydrothermal liquor. The quantification of produced XOS was performed by HPLC, and XOS yield was determined for each cycle. As shown in Figure 7, Purolite-MpXyn10 remained stable in six successive cycles of enzymatic hydrolysis (hydrolysis-washing-hydrolysis) for both substrates, giving more than 80% conversion after 3 h at 50 °C (Figure 7).

## 3. Materials and Methods

### 3.1. Reagents, Materials, and Raw Materials

Agarose 6 BCL was purchased from GE Healthcare Bio-Sciences AB (Uppsala, Sweden). Epichlorohydrin, iminodiacetic acid, ethylenediamine, polyethyleneimine, 3,5-dinitrosalicilic acid (DNS), orcinol, *beechwood* xylan (~90%), *birchwood* xylan (~90%), oat spelts xylan (~75%), xylose (>95%), arabinose (>95%), glucose (>95%), furfural (99%), and 5-(Hydroxymethyl) furfural, HMF (99%) were purchased from Sigma-Aldrich (St. Louis, MO, USA). Wheat Arabinoxylan (~95%), xylobiose (>95%), xylotriose (>95%), xylotetraose (>95%), xylopentaose (>95%), and xylohexaose (>95%) were acquired from Megazyme (Bray, Wicklow, Ireland). Silica gel plates DC-Alufolien Kieselgel 60 were purchased from Fluka (Darmstadt, Germany). Bradford protein assay and Precision Plus protein TM standards were purchased from Bio-Rad Laboratories (Hercules, CA, USA). Hiprep Q FF and Superdex 75 10/300 GL columns were acquired from GE Healthcare (Uppsala, Sweden). Chromatography HPLC-column Rezex™ ROA-Organic Acid H+ was purchased from Phenomenex (Torrance, CA, USA). Amino C2 methacrylate beads (Purolite) support was kindly provided by Purolite S.A (Victoria, Romania). All other reagents used for the assays were of analytical grade.

*Eucalyptus grandis* wood chips were kindly donated by the International Paper Company (Luiz Antônio, SP, Brazil). Before use, the raw material was dried at 50 °C and milled in a 25-mesh sieve knife mill (SL-32–SOLAB).

### 3.2. Fungal Strain

*Aspergillus nidulans* A773 recombinant strain carrying the construction pEXPYR/Mpxyn10 was used to produce the endo-1,4-β-xylanase from *Malbranchea pulchella* (Mpxyn10) [15]. The strain was maintained at 4 °C on complete medium (CM) slants. Before use, it was inoculated onto CM and incubated at 37 °C for three days for sporulation. The solution of the spores was prepared with sterile distilled water, and the spore concentration in the suspension was determined using a Neubauer chamber.

### 3.3. Production and Purification of Recombinant MpXyn10 Xylanase

Production of MpXyn10 was carried out in a 5 L stirred tank bioreactor BioFlo^®^/CelliGen^®^ 310 (New Brunswick/Eppendorf, Hamburg, Germany). Briefly, 20 mL of a spore suspension (1 × 108 spores mL^−1^) of *A. nidulans* A773 was inoculated in 500 mL of minimal medium, pH 6.5 [58], supplemented with 5% corn syrup (Liquid Glucose, Marvi, Ourinhos, SP, Brazil). Pre-inoculum was incubated for 48 h at 37 °C and 150 rpm. Afterward, batch fermentation was carried out with the same culture medium (workload of 4.5 L in the reactor vessel), at 37 °C, with aeration of 1.0 v.v.m L^−1^ and 300 rpm. After 60 h of cultivation, the crude extract was filtered in a Büchner funnel with Whatman n° 1 filter paper and concentrated using the QuixStand Benchtop Systems tangential filtration concentrator with a 10 kDa Hollow Fiber Cartridge (GE Healthcare, Chicago, IL, USA). The concentrated crude extract was loaded on an anion exchange column (Hiprep Q FF 16/10) equilibrated with 50 mmol L^−1^ phosphate buffer, pH 7.5, integrated with an ÄKTA Purifier 900 (GE Healthcare Life Science, Cranbury, NJ, USA) chromatography system. Protein elution was monitored at 280 nm, with a linear gradient from 0 to 1 mol L^−1^ of NaCl. The protein samples were analyzed using SDS-PAGE and selected, pooled, and concentrated using Vivaspin ultra-filtration devices with a 10 kDa membrane (Sigma-Aldrich, St. Louis, MO, USA). The recovered piece was loaded in a size exclusion column (Superdex 75 10/300 GL), equilibrated with phosphate buffer 50 mmol L^−1^, pH 7.0, plus NaCl 150 mmol L^−1^. Protein elution was monitored at 280 nm, and the fractions of interest were selected as described above.

### 3.4. Electrophoresis and Protein Content Determination

Electrophoresis SDS-PAGE (15%) was performed according to Laemmli [59], using a molecular standard of 10 to 250 kDa (Protein TM Standards—Bio-Rad, Hercules, CA, USA). Protein content was determined through the method of Bradford [60], using bovine serum albumin as the standard. Gel samples were stained with a Coomassie Brilliant Blue R-250 solution (0.05% *m*/*v*) and distained with a methanol/acetic acid/water solution (5/1/4 *v*/*v*/*v*).

### 3.5. Xylanase Activity Assay

Xylanase activity was determined with the reducing-sugars method using 3,5 dinitrosalicylic acid (DNS) [61]. The assays were carried out with 50 µL of enzyme soluble or immobilized form (0.0015 g), 50 µL of citrate buffer (0.1 mol L^−1^, pH 5.0), and 100 µL of 1% xylan *beechwood* (*w*/*v*). The assay was incubated in a Thermomixer comfort (Eppendorf, Hamburg, Germany) for 10 min at 50 °C and 1300 rpm. After incubation, 200 µL of DNS were added to the assay mixture and boiled at 98 °C for 5 min. Afterward, 1 mL of distilled water was added to the mix, and 100 μL samples were collected and analyzed for reducing sugar at 540 nm using microplate readers (Spectramax M2, Molecular Devices, San Jose, CA, USA). A standard curve of D-xylose was used to estimate the reducing sugar concentrations equivalents. One unit of enzyme activity (U) was defined as the amount of enzyme that catalyzes the release of 1 μmoL of reducing sugar per minute under the assay conditions.

### 3.6. Preparation and Activation of Supports

Epoxy-agarose support was prepared according to [15,57]. First, Agarose 6BL (25 g) was mixed with 110 mL of distilled water, 8.2 g of NaOH, 0.5 g of NaBH_4_, and 40 mL of acetone. The mixture was kept in suspension by stirring with paddles in an ice bath, and 27.5 mL of epichlorohydrin was added dropwise. Afterward, the suspension was gently stirred for 16 h at 25 °C. Finally, epoxy-activated agarose was washed with distilled water, filtered using a glass filter, and stored at 4 °C.

Glyoxyl-agarose support was prepared as previously described [43,45]. First, epoxy-activated agarose (20 g) was mixed with 200 mL of 1 mol L^−1^ H_2_SO_4_ and gently stirred for 2 h at 25 °C. Then, the support was washed with distilled water, filtered, and resuspended in 200 mL of 100 mmol L^−1^ NaIO_4_ for 2 h at 25 °C. Afterward, the glyoxyl-agarose support was washed with the excess distilled water, filtered, and stored at 4 °C.

Monoaminoethyl-N-aminoethyl (MANAE)-agarose support was prepared according to Fernández-Lafuente et al. [40]. Briefly, 10 g of glyoxyl-agarose was mixed with 100 mL of 2 mol L^−1^ ethylenediamine (EDA) at pH 10.5. The suspension was gently stirred for 2 h at 25 °C. Afterward, NaBH_4_ (10 mg mL^−1^) was added to the mixture, which was kept under gentle stirring for 2 h at 25 °C. The support was then washed with 1 L of 100 mmol L^−1^ sodium acetate, pH 4.0, and 1 L of 100 mmol L^−1^ sodium borate, pH 9.0. Finally, MANAE-agarose support was washed with the excess distilled water, filtered using a glass filter, and stored [34,62] at 4 °C.

Agarose-activated with glutaraldehyde groups (GLUT-agarose) was prepared as previously described by [34,63]. MANAE-agarose support (5 g) was mixed with 6 mL of glutaraldehyde solution (25%, *v*/*v*) and added in 9 mL of 200 mmol L^−1^ of phosphate buffer, pH 7.0. The mixture was kept under gentle stirring for 18 h at 25 °C. Afterward, GLUT-agarose support was washed with the excess distilled water, filtered using a glass filter, and stored at 4 °C.

PEI-agarose support was prepared according to [64]. Glyoxyl-agarose (10 g) was added to 90 mL of 100 mmol L^−1^ sodium bicarbonate buffer, pH 11, containing 10% (*w*/*v*) of polyethyleneimine (PEI). The mixture was kept gently stirred for 3 h at 25 °C. Afterward, 1 g of NaBH_4_ was added, and the mixture was incubated for a further 2 h at 25 °C. PEI-activated agarose was washed with distilled water, filtered using a glass filter, and stored at 4 °C.

### 3.7. Fourier Transform Infrared (FTIR)

The chemical groups on the untreated and treated support were identified by FTIR using a Perkin Elmer model Frontier spectrophotometer equipped with a diamond ATR module (Perkin Elmer, Waltham, MA, USA). Measurements were taken in the range of 200 to 4000 cm^−1^ with a resolution of ±1 cm^−1^ and 120 scans. FTIR spectra were not used as a quantitative relationship but as a qualitative reference.

### 3.8. Immobilization Assays

Immobilization assays were carried out using a standard protocol for all supports. Briefly, 1 g of support was suspended in 10 mL of a solution containing an enzyme loading of 25 U per g of support (~0.2–0.3 mg of protein loading). The immobilization was performed under mild stirring (80 rpm) at 25 °C. For MANAE-agarose, PEI-agarose, and Purolite support, the enzyme was added to a phosphate buffer at 10 mmol L^−1^, pH 7.0. For Glyoxyl-agarose, sodium bicarbonate buffer 100 mmol L^−1^, pH 10.5, was used, and after the incubation, it was reduced by adding NaBH_4_ (1 mg mL^−1^) for 30 min. The time course of immobilization was evaluated by determining the residual activity in aliquots of the supernatant and suspension removed over time. The parameters of the immobilization were defined as the immobilization yield (IY), calculated according to Equation (1):(1)$$\mathrm{IY}=\frac{\left(\mathrm{Uti}\;–\;\mathrm{Utf}\right)\;}{\mathrm{Uti}}\times 100,$$
where Uti is the activity (U mL^−1^) of the supernatant before immobilization and Utf is the activity (U) remaining in the supernatant at the end of the immobilization procedure.

The activity recovery (AR) was calculated according to Equation (2):(2)$$\mathrm{AR}=\frac{{\mathrm{UI}}_{o}\;}{\mathrm{UT}}\times 100$$
where UI_o_ is the observed activity of the immobilized derivative (U g^−1^ of support) and UT is the theoretical activity of the immobilized preparation (U g^−1^ of support), calculated based on the difference between the initial activity and the removed activity from the supernatant during the immobilization procedure.

### 3.9. Characterization of the MpXyn10 Immobilized Derivatives

#### 3.9.1. Stability at pH and Temperature

Thermal stabilities of free and immobilized MpXyn10 were assessed with incubation in 50 mmol L^−1^ citrate buffer, pH 5.0, at 65 and 70 °C. For pH stability assays, free and immobilized MpXyn10 was incubated in a 50 mmol L^−1^ citrate buffer in the pH range of pH 4.0–6.0 for 24 h at 25 °C. Aliquots were withdrawn periodically to quantify residual enzymatic activity as described in 2.4. The residual activities for both experiments were calculated concerning controls treated identically but without incubation.

#### 3.9.2. Effect of Furfural and 5-Hydroxymethyl-Furfural on MpXyn10 Activity

The effect of furfural and 5-hydroxymethyl-furfural compounds on MpXyn10 activity was determined by adding furfural (0.48 g L^−1^) and HMF (0.63 g L^−1^) in the reaction medium corresponding to 5 mmol L^−1^. The relative activities for free and immobilized MpXyn10 were calculated concerning controls treated identically but without adding the compounds.

### 3.10. Enzymatic Hydrolysis of Xylan

Free- and immobilized MpXyn10 were applied in the hydrolysis of xylans from diverse sources (xylan from *beechwood*, *birchwood*, oat spelts, and arabinoxylan). The assays consisted of 0.5% (*w*/*v*) xylan in 50 mmol L^−1^ citrate buffer, pH 5.0, and 0.1 U of xylanase activity in the reaction medium for both preparations. The assays were incubated in a Thermomixer comfort (Eppendorf, Hamburg, Germany) at 50 °C, 1300 rpm. Aliquots were removed over time (0 to 180 min) and boiled at 98 °C for 5 min. The reducing sugars were quantified with the DNS method, and the XOS produced were analyzed with TLC and HPLC. Controls were carried out to evaluate the xylan-derived products in the reaction medium under the same assay conditions.

The XOS yield, % (xylan conversion to xylobiose (X2) and xylotriose (X3) was calculated according to Equation (3):(3)$$\mathrm{XOS}\;\mathrm{Yield}=\frac{\left(f_{X2}\times {\;C}_{X2}\right)+\left(f_{X3}\times {\;C}_{X3}\right)\;}{\mathrm{Initial}\;\mathrm{xylan}\;\left({\mathrm{mg}\;\mathrm{mL}}^{-1}\right)}\times 100$$
where C is the concentration (mg mL^−1^); f_X2_ (0.936) and f_X3_ (0.956) are stoichiometric factors [21,53,65].

### 3.11. Liquid Hot Water (LHW) Pretreatment and Compositional Analysis of the Eucalyptus Biomass

The LHW pretreatment of *Eucalyptus grandis* wood chips (EWC) was carried out according to Silva et al. [63]. Briefly, the assays were performed in a 250-mL stainless steel cylinder reactor (5.0 cm internal diameter × 12.5 cm internal height), with a working volume of 30 mL, at 3% (*w*/*v*) of solid loading. The reactor was immersed in an oil bath (QUIMIS model Q213, Diadema, SP, Brazil) at 160 °C for 60 min (severity factor = 3.54). These conditions were used to obtain a better recovery of the xylan-rich liquor after the pretreatment avoiding the formation of unwanted products, such as xylose, furfural, and HMF [66,67]. Afterward, the reactor was immediately cooled in an ice bath to stop the reaction. The hydrothermal liquor was separated from the insoluble solids through filtration with qualitative filter paper (Whatman n° 1) and stored at −20 °C until use.

The chemical composition of in natura and pretreated EWC was determined according to the US National Renewable Energy Laboratory (NREL) [68,69]. The analyzed components were glucan, xylan, arabinan, acetyl groups, Klason lignin, and ashes. Hemicellulose-derived products (XOS, xylose, glucose, arabinose, and acetic acid), as well as furfural and HMF of hydrothermal liquor, were determined before and after acid hydrolysis with 4% (*v*/*v*) sulfuric acid, at 121 °C, for 20 min [47]. HPLC analysis was performed using a Metacarb 87H carbohydrate analysis column 300 × 7.8 mm (Varian, Palo Alto, CA, USA) at 60 °C. Sugars and acetic acid were analyzed with a refractive index (RI) detector, and furfural and HMF with a UV detector, both in a Jasco chromatograph. The mobile phase was 0.005 M H_2_SO_4_ in ultrapure water, and the flow rate was 0.7 mL/min.

### 3.12. Production of XOS from EWC Hydrothermal Liquor

After LHW pretreatment, enzymatic hydrolysis of xylan present in EWC liquor was carried out by employing free or immobilized MpXyn10. Initially, the reaction medium consisted of 1.5 mL containing EWC liquor in 50 mmol L^−1^ citrate buffer (50% *v*/*v*), pH 5.0. The assay was carried out as described in item 3.10. For the operational stability of the immobilized MpXyn10 derivatives, EWC liquor was used as the substrate to produce the low DP XOS. Immobilized preparations (2 U) were added in a 2-mL vial containing the reaction medium (1 mL) at 50 °C, 150 rpm. After each reaction cycle, the immobilized preparations were recovered with filtration using a glass filter and washed thrice with 50 mmol L^−1^ citrate buffer, pH 5.0, and then reused on a new reaction cycle. XOS yield was determined for each cycle.

### 3.13. Analysis of the Xylan Hydrolysis Products by Thin-Layer Chromatography

Thin-layer chromatography (TLC) analyses were performed on silica gel plates (10 × 10 cm) (Fluka, Darmstadt, Germany) at room temperature using a mobile phase composed of ethyl acetate, acetic acid, formic acid, and water in a ratio of 9:3:1:4 (*v*/*v*) [70,71]. Before the runs, the samples from 0, 30, 90, and 180 min (described in Section 3.10) were filtered in a 0.22-μm filter, precipitated overnight with 10% of trichloroacetic acid, and centrifuged at 12,000 × *g* for 10 min. Then, ten microliters of each supernatant were applied to TLC plates. After two runs with the mobile phase, the plates were pulverized with a solution composed of orcinol 0.04% (*w*/*v*) dissolved in ethanol/sulfuric acid (9:1, *v*/*v*) and heated at 150 °C until the appearance of dark spots. Xylose (X1), xylobiose (X2), xylotriose (X3), xylotetraose (X4), xylopentose (X5), xylohexose (X6) were used as TLC standards.

### 3.14. HPLC Analysis

XOS and sugar profiles of hydrothermal liquor were determined using a High-Performance Liquid Chromatography system (YL9100 model, Lin Instruments, Anyang, Gyeonggi, Korea) equipped with a YL9170 refractive index detector and acid H^+^ column Rezex™ ROA-Organic (8%) (300 × 7.8 mm diameter), at 80 °C. Samples were filtered in a 0.22-μm filter and injected using 0.005 mmol L^−1^ H_2_SO_4_ as the mobile phase at a flow rate of 0.5 mL min^−1^ and a run time of 25 min. Standards were dissolved in the mobile phase at different concentrations and analyzed under the same conditions of samples. XOS were quantified using average peak areas compared with a mixture of standards, xylobiose (X2) and xylotriose (X3), and expressed in mg mL^−1^.

## 4. Conclusions

In the current work, we successfully immobilized and applied the recombinant MpXyn10 xylanase to produce XOS from different xylan sources. The immobilization process does not alter the biochemical properties of MpXyn10 and the hydrolysis patterns. Besides, it allows the obtention of biocatalysts with high activity and thermal- and operational stability, critical features for application in bioprocesses. Our results also show that the liquor produced from hydrothermal pretreatment of the eucalyptus wood chips residues could be used as a source to produce XOS. The liquor, rich in soluble xylan, was used as a substrate in the efficient production of XOS by Purolite-MpXyn10 in various hydrolysis cycles, which shows the potential of the immobilized recombinant xylanase MpXyn10 in integrated bioprocess in a biorefinery for complete conversion of lignocellulosic materials. The results obtained in this work provide a basis for applying MpXyn10 in the hydrolysis of xylan from EWC hydrothermal liquor but also open the possibility of its application (together with cellulases) in pretreated solids, even for other LCM.

## Figures and Tables

**Figure 1 ijms-23-13329-f001:**
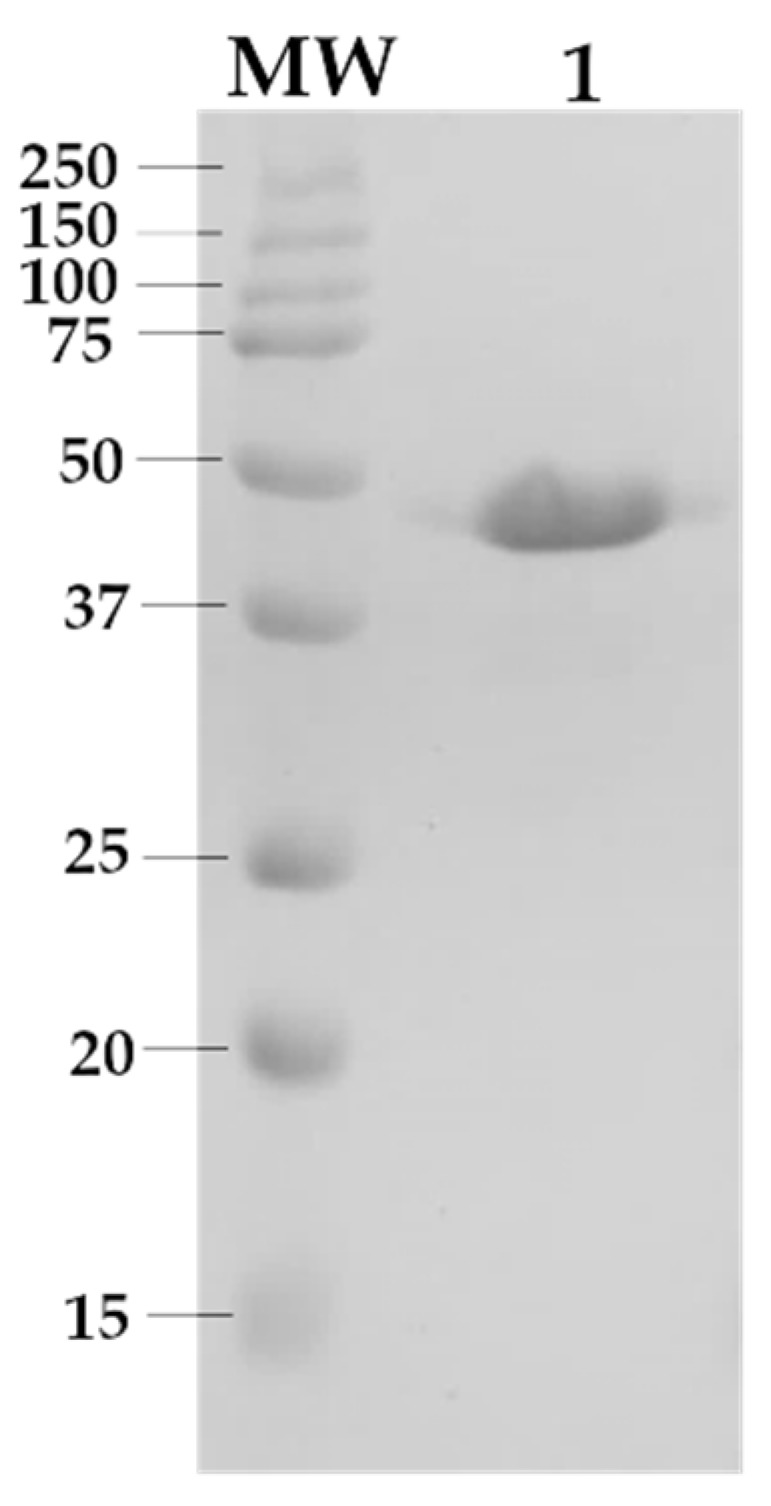
Purification of the recombinant endo-1,4-β-xylanase from *M. pulchella* (MpXyn10) expressed in *A. nidulans* A773. Gel SDS-PAGE (15%). MW: molecular weight standard ladder shown as kDa. Lane 01: Fraction eluted from Superdex 75 10/300 GL size exclusion chromatography column. Protein samples were stained with Coomassie Blue R-250.

**Figure 2 ijms-23-13329-f002:**
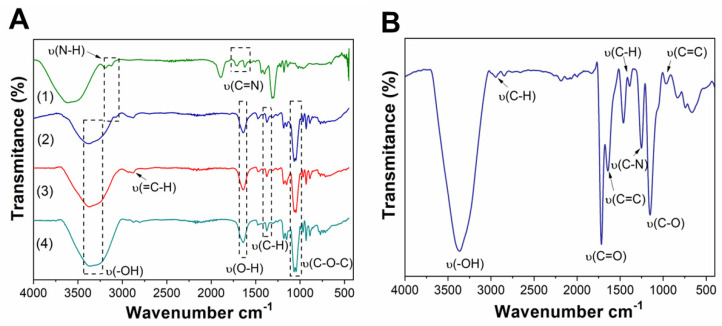
(**A**) Fourier Transformed Infrared (FTIR) spectra of the agarose-activated supports: (1) GLUT-agarose; (2) MANAE-agarose; (3) Glyoxyl-agarose, and (4) Agarose beads. (**B**) FTIR of the amino C2 methacrylate beads (Purolite) support.

**Figure 3 ijms-23-13329-f003:**
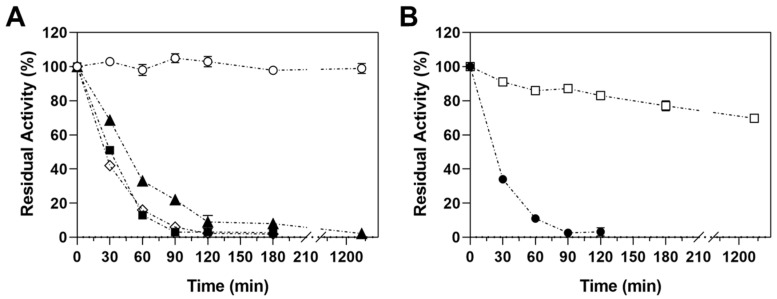
Immobilization of MpXyn10 on different supports. (**A**) Immobilization courses of MpXyn10 on amino-activated supports and (**B**) aldehyde-activated supports. Key: Control (○), PEI-agarose (▲), MANAE-agarose (■), Purolite (◊), GLUT-agarose (●), and Glyoxyl-agarose (□). The error bars represent the standard error of the mean.

**Figure 4 ijms-23-13329-f004:**
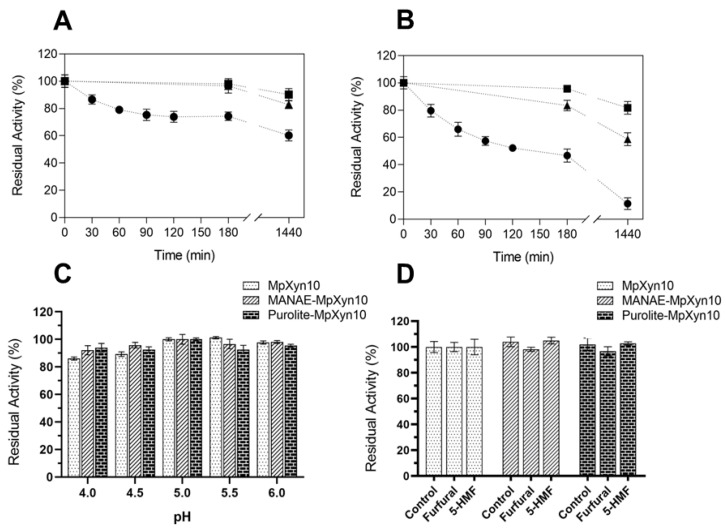
Characterization of the immobilized MpXyn10. Thermal stability (**A**) at 65 °C and (**B**) at 70 °C. Key: Free MpXyn10 (●); MANAE-MpXyn10 (■) and Purolite-MpXyn10 (▲). (**C**) Stability at pH of free and immobilized MpXyn10. The residual activity was determined after incubation for 24 h at 25 °C in different citrate buffers (pH 4.0–6.0, 50 mmol L^−1^). (**D**) Effect of furfural and HMF on the activity of free and immobilized MpXyn10. The relative activity was determined as described in 2.4. The error bars represent the standard error of the mean.

**Figure 5 ijms-23-13329-f005:**
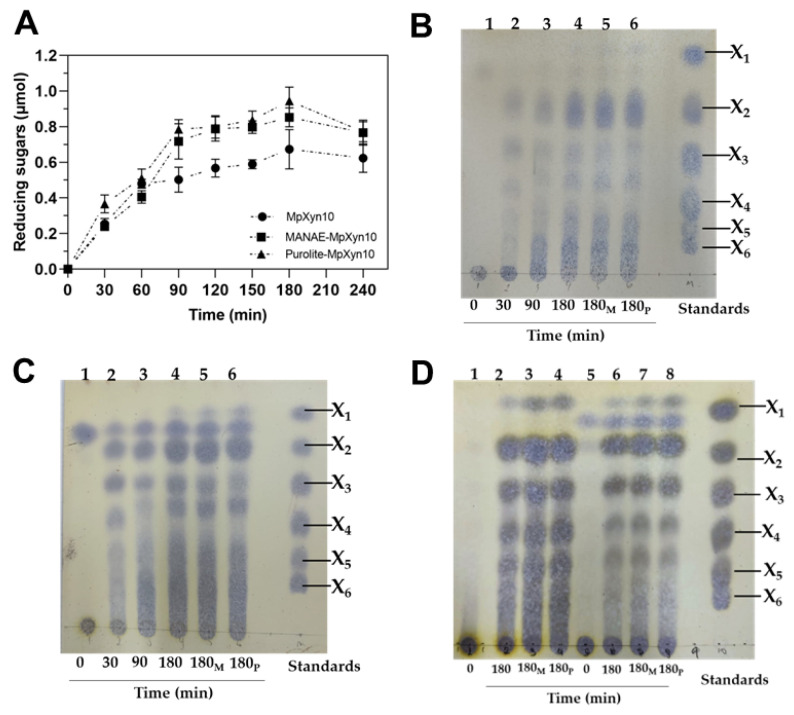
Profile of xylan hydrolysis from free and immobilized MpXyn10. (**A**) Reducing sugars released from *beechwood* xylan hydrolysis catalyzed with MpXyn10. (**B**) Thin Layer Chromatography (TLC) profile of hydrolysis of *beechwood* xylan, (**C**) wheat arabinoxylan, (**D**) *birchwood* and oat spelt xylans. The reducing sugars were quantified with the DNS method. Key: (**A**) free MpXyn10 (●), MANAE-MpXyn10 (■), and Purolite-MpXyn10 (▲); (**B**,**C**) lane 1: the beginning of the reaction; lanes 2–4: free MpXyn10; lane 5: MANAE-MpXyn10; lane 6: Purolite-MpXyn10; (**D**) lanes 1 and 5: the beginning of the hydrolysis of xylan *birchwood* and oat spelts, respectively; lanes 2 and 6: free MpXyn10; lanes 3 and 7: MANAE-MpXyn10; lanes 4 and 8: Purolite-MpXyn10. Controls were carried out to evaluate the spontaneous hydrolysis of the substrate under the same assay conditions.

**Figure 6 ijms-23-13329-f006:**
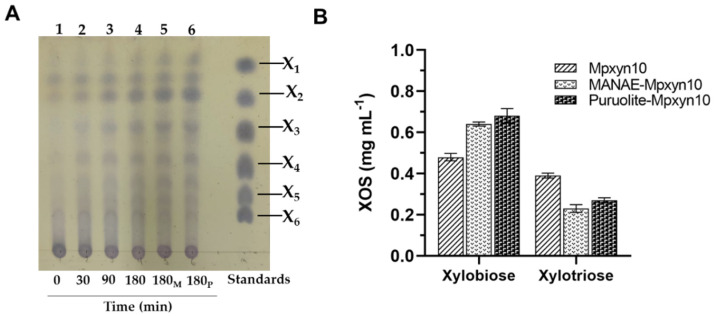
Production of XOS from soluble xylan of the EWC hydrothermal liquor. (**A**) Thin Layer chromatography (TLC) of the enzymatic hydrolysis products of EWC hydrothermal liquor. Lane 1: the beginning of the assay (control); lanes 2–4: free MpXyn10; lane 5: MANAE-MpXyn10; lane 6: Purolite-MpXyn10. (**B**) Quantification of XOS (xylobiose and xylotriose) produced after 180 min of the assay by HPLC.

**Figure 7 ijms-23-13329-f007:**
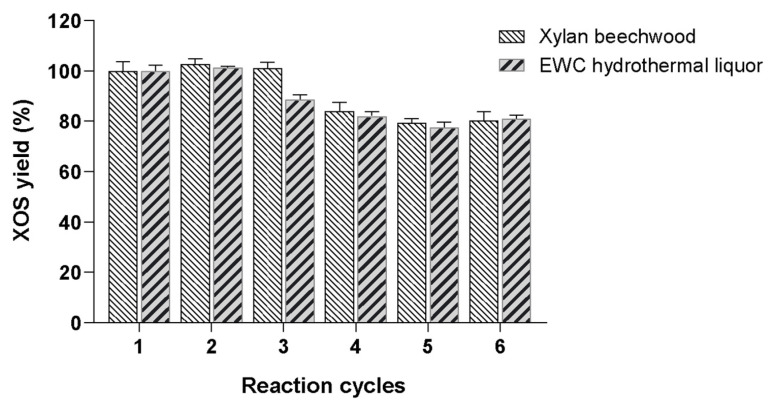
Operational stability of Purolite-MpXyn10 during successive reaction cycles for XOS production. Purolite-MpXyn10 derivative was used to catalyze six successive hydrolysis reactions of xylan *beechwood* and EWC hydrothermal liquor. The relative production of XOS was expressed as percentages of the products obtained in the first reaction cycle.

**Table 1 ijms-23-13329-t001:** Principal parameters for immobilization of the Mpxyn10 on different supports.

Support	Immobilization Yield ^1^ (%), IY	Activity (U g^−1^ Support)	Activity Recovery ^2^ (%), AR
Glyoxyl-agarose	30	3.8 ± 0.9	51
MANAE-agarose	>95	31 ± 1.0	137
GLUT-agarose	>95	15.5 ± 0.6	61
PEI-agarose	>95	27.7 ± 1.3	105
Purolite	>95	35.6 ± 2.1	142

^1^ Calculated as the difference between the initial and final activities (U mL^−1^) in the supernatant after immobilization. ^2^ Activity recovery (%), measured as the ratio between the observed activity (U g^−1^ support) of immobilized MpXyn10 and theoretical activity of the immobilized MpXyn10 (U g^−1^ support). The ± represents the standard error of the mean.

**Table 2 ijms-23-13329-t002:** Products obtained from the hydrolysis of the different xylans catalyzed by free and immobilized MpXyn10.

Substrate	Biocatalyst	Hydrolysis Products ^1^ (mg mL^−1^)			Yield XOS (%)
		Xylose	Xylobiose	Xylotriose	
Xylan *beechwood*	MpXyn10	0.037 ± 0.01	6.11 ± 0.14	Nd	38
	MANAE-MpXyn10	0.040 ± 0.01	7.88 ± 0.20	0.21 ± 0.02	50
	Purolite-MpXyn10	0.075 ± 0.03	7.78 ± 0.06	0.20 ± 0.01	50
Xylan *birchwood*	MpXyn10	0.040 ± 0.01	5.66 ± 0.48	0.46 ± 0.03	38
	MANAE-MpXyn10	0.042 ± 0.01	7.46 ± 0.42	0.60 ± 0.07	50
	Purolite-MpXyn10	0.081 ± 0.03	7.60 ± 0.50	0.77 ± 0.11	52
Xylan oat spelts	MpXyn10	0.031 ± 0.01	3.64 ± 0.11	0.50 ± 0.01	26
	MANAE-MpXyn10	0.070 ± 0.01	6.64 ± 0.03	0.91 ± 0.01	47
	Purolite-MpXyn10	0.106 ± 0.01	6.91 ± 0.07	0.99 ± 0.03	49
Wheat arabinoxylan	MpXyn10	0.174 ± 0.02	6.34 ± 0.52	0.59 ± 0.07	43
	MANAE-MpXyn10	0.106 ± 0.01	6.36 ± 0.48	0.61 ± 0.02	44
	Purolite-MpXyn10	0.806 ± 0.10	7.79 ± 0.24	1.14 ± 0.05	56

^1^ Xylooligosaccharides were determined using HPLC as described in item 3.14. Nd: Not detected. The ± represents the standard deviation.

**Table 3 ijms-23-13329-t003:** Chemical composition of raw and pretreated *Eucalyptus grandis* wood chips (EWC), expressed as a percentage on a dry weight basis and reported as average values (±standard deviation).

Components	Raw EWC	LHW-Pretreated EWC
**Solids (%, *w*/*w*)**		
Cellulose ^1^	33.43 ± 3.08	40.85 ± 2.67
**Hemicellulose**		
Xylan	14.82 ± 1.50	9.95 ± 1.03
Arabinan	0.58 ± 0.01	Nd
Acetyl group	1.59 ± 0.26	Nd
Klason lignin	27.76 ± 1.63	30.81 ± 1.76
Ashes	9.93 ± 2.45	-
Others ^2^	13.48	18.39
**Hydrolysates (g/L)**		
**Oligosaccharides**		
Glucooligosaccharides		Nd
Xylooligosaccharides		2.26 ± 0.19
Arabinooligosaccharides		Nd
Acetyl groups-oligosaccharides		Nd
**Monosaccharides**		
Glucose		0.171 ± 0.00
Xylose		0.294 ± 0.12
Arabinose		0.308 ± 0.09
Acetic acid		0.183 ± 0.01
**Degradation products**		
Hydroxymethylfurfural		0.010 ± 0.00
Furfural		0.234 ± 0.01

^1^ Estimated glucan contents. ^2^ Calculated by difference (includes non-analyzed components). Nd: not detected. The ± represents the standard deviation.

## Data Availability

Not applicable.

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
