# Peer review of "Immobilization and Application of the Recombinant Xylanase GH10 of Malbranchea pulchella in the Production of Xylooligosaccharides from Hydrothermal Liquor of the Eucalyptus (Eucalyptus grandis) Wood Chips"

_ijms, 2022, doi:10.3390/ijms232113329_

Round 1
Reviewer 1 Report
The article «Immobilization and application of the recombinant xylanase GH10 of Malbranchea pulchella in the production of xylooligosaccharides from hydrothermal liquor of the eucalyptus (Eucalyptus grandis) wood chips» has a high applied value. The scheme of the experiment is quite understandable and reproducible.
A recombinant endo-1,4-β-xylanase of Malbranchea pulchella (Mpxyn10) was immobilized in various chemical supports. MpXyn10 derivatives were tested to produce XOS from xylan of various sources. The results obtained provide a basis for the application of immobilized MpXyn10 to produce XOS and other high value-added products in the lignocellulosic biorefinery field.
Major revision:
1. In Figure 1, the sample MpXyn10 has a wide band. Authors either added too much of the sample, or it was inhomogeneous. Authors should explain this.
2. It is not clear from Table 1 what kind of activity is meant – total or specific. Was the protein content of the samples taken into account?
Page 14: authors, write “One unit of enzyme activity (U) was defined as the amount of enzyme that catalyzes the release of 1 μmoL of reducing sugar per minute under the assay conditions”. It is necessary to indicate how they measured the amount of enzyme. Data on the amount of enzyme is nowhere in the article. It is appropriate to add them to the Table. 1.
Activity and immobilization yield are very different for Glyoxyl-agarose, MANAE-agarose, GLUT-agarose, PEI-agarose, Purolite. The authors should explain these effects (at least according to the literature), add the chemical formulas of the compounds, describe the features and mechanisms of immobilization on them.
Minor revisions:
1. Authors are fond of self-quoting: 13 references to the works of authors out of 68, that is, about 20%.
2. Page 3, fifth line from top: different font “XOS are sugar oligomers composed of xylose units linked by β-(1→4) bonds”.
3. Authors make inaccuracies in the description of the methods: Page 14, second line from top: they write “The assays were carried out with 50 μL of enzyme sample (free or immobilized form)”. This is not entirely correct, because Purolite is insoluble.
Author Response
Ribeirão Preto, October 24th, 2022.
Dear Dr. Karl Wu
International Journal of Molecular Sciences, Editor
Manuscript Number: IJMS-1972220
We are submitting a revised version of the manuscript entitled " Immobilization and application of the recombinant xylanase GH10 of Malbranchea pulchella in the production of xylooligosaccharides from hydrothermal liquor of the eucalyptus (Eucalyptus grandis) wood chips". We appreciate the attention and the suggestions received to improve our research, and we would like to thank the reviewers for investing their valuable time in reviewing our manuscript. The answers to all questions are listed below. We list the reviewers' comments by numbering them according to their sequence in the reviewers' questions and their respective answers for better understanding. We have revised the entire manuscript and highlighted the changes in red. We have also read carefully through the text and corrected any other minor mistakes we have found.
I look forward to hearing from you.
Dr. Maria de Lourdes T. M. Polizeli
RESPONSES TO REVIEWERS
Reviewer #1:
The article «Immobilization and application of the recombinant xylanase GH10 of Malbranchea pulchella in the production of xylooligosaccharides from hydrothermal liquor of the eucalyptus (Eucalyptus grandis) wood chips» has a high applied value. The scheme of the experiment is quite understandable and reproducible. A recombinant endo-1,4-β-xylanase of Malbranchea pulchella (Mpxyn10) was immobilized in various chemical supports. MpXyn10 derivatives were tested to produce XOS from xylan of various sources. The results obtained provide a basis for the application of immobilized MpXyn10 to produce XOS and other high value-added products in the lignocellulosic biorefinery field.
- In Figure 1, the sample MpXyn10 has a wide band. The authors either added too much of the sample, or it was inhomogeneous. The authors should explain this.
Thank you. The reviewer is undoubtedly right about a wide band in SDS-PAGE. As the reviewer says, it is the amount of protein sample added. As the original SDS-PAGE below (figure a) shows, the band corresponding to MpXyn10 was observed in the purified fraction. To comply with the reviewer's request and make the SDS-PAGE clearer, we improved the resolution of Figure 1 (figure b) in the revised manuscript.
- (b)
- It is not clear from Table 1 what kind of activity is meant – total or specific. Was the protein content of the samples taken into account? Page 14: authors, write "One unit of enzyme activity (U) was defined as the amount of enzyme that catalyzes the release of 1 μmoL of reducing sugar per minute under the assay conditions". It is necessary to indicate how they measured the amount of enzyme. Data on the amount of enzyme is nowhere in the article. It is appropriate to add them to the Table. 1.
Thank you for the valuable suggestion. We comply with the reviewer's request. The immobilization yield (IY) was calculated from activity in U mL-1 of the supernatant (Eq.1), and activity recovery (AR) was calculated using total activity. The protein content was not taken into account. Thus, to make the text clearer, we improved the footnote of Table 1 and addressed the protein amount used of the purified MpXyl10 in the Materials and methods section of the revised manuscript, as requested.
- Activity and immobilization yield are very different for Glyoxyl-agarose, MANAE-agarose, GLUT-agarose, PEI-agarose, Purolite. The authors should explain these effects (at least according to the literature), add the chemical formulas of the compounds, describe the features and mechanisms of immobilization on them.
Thank you for the valuable suggestion. In response to this comment, we addressed most of the points raised by the reviewer. Please, see lines 148 to 181 of the revised manuscript:
- Authors are fond of self-quoting: 13 references to the works of authors out of 68, that is, about 20%.
We thank you and agree with the comment. However, we realize that references in the manuscript follow their relevance within the subject discussed, mainly in the case of previous works of our research group. Thus, to comply with the reviewer's request and improve the raised point, we revised the references and sought to update them according to their relevance in each subject. Please see the references changed in the revised manuscript.
- Page 3, fifth line from top: different font "XOS are sugar oligomers composed of xylose units linked by β-(1→4) bonds"
We comply with the reviewer's request. These typographical errors were corrected in the revised manuscript. We also revised the manuscript carefully and corrected any other minor mistakes we found.
- Authors make inaccuracies in the description of the methods: Page 14, second line from top: they write "The assays were carried out with 50 μL of enzyme sample (free or immobilized form)". This is not entirely correct, because Purolite is insoluble.
We revised and improved the description in the Materials and methods section of the revised manuscript, as requested.

Reviewer 2 Report
Alnoch et al. report on the recombinant production, purification, immobilization, and application of the thermotolerant Malbranchea pulchella xylanase in the production of xylooligosaccharides (XOS), sugar oligomers composed of xylose units linked by beta – 1,4 linkages. Xylan, the most common hemicellulosic polysaccharide in the plant cell wall and the second most abundant polysaccharide in nature, is primarily constituted by xylose monomers joined by beta-1,4 glycosidic linkages. Endo-1,4-beta-xylanses are the main enzymes in cleaving beta-1,4 glycosidic bonds from xylan and releasing XOS of varied sizes. XOS, which has a number of important uses in food and in medicine, is largely produced by chemical hydrolysis. More recent research has led to the use of free and immobilized xylanases. This manuscript details the use of a fungal MpXyn10 xylanase recombinantly produced and purified from Aspergillus nidulans and then immobilized on a number of different supports. The researchers found that the MpXyn immobilized on Purolite and on MANAE-agarose retained a significant amount of activity, thermostability, and pH stability when compared to the free enzyme. In addition, the profile of products, typically xylobiose and xylotriose, was consistent across supports, and the XOS yield was ~50% for both the Purolite and MANAE-agarose immobilized enzymes.
Specific Comments:
(1) A few grammatical and typographical errors need to be resolved.
(2) The abstract could be strengthened by indicating the significance of the research detailed in the manuscript. It gets lost why the average reader should be concerned with the production of the XOS by immobilized xylanases.
Author Response
Ribeirão Preto, October 24th, 2022.
Dear Dr. Karl Wu
International Journal of Molecular Sciences, Editor
Manuscript Number: IJMS-1972220
We are submitting a revised version of the manuscript entitled " Immobilization and application of the recombinant xylanase GH10 of Malbranchea pulchella in the production of xylooligosaccharides from hydrothermal liquor of the eucalyptus (Eucalyptus grandis) wood chips". We appreciate the attention and the suggestions received to improve our research, and we would like to thank the reviewers for investing their valuable time in reviewing our manuscript. The answers to all questions are listed below. We list the reviewers' comments by numbering them according to their sequence in the reviewers' questions and their respective answers for better understanding. We have revised the entire manuscript and highlighted the changes in red. We have also read carefully through the text and corrected any other minor mistakes we have found.
I look forward to hearing from you.
Dr. Maria de Lourdes T. M. Polizeli
RESPONSES TO REVIEWERS
Reviewer #2:
Alnoch et al. report on the recombinant production, purification, immobilization, and application of the thermotolerant Malbranchea pulchella xylanase in the production of xylooligosaccharides (XOS), sugar oligomers composed of xylose units linked by beta – 1,4 linkages. Xylan, the most common hemicellulosic polysaccharide in the plant cell wall and the second most abundant polysaccharide in nature, is primarily constituted by xylose monomers joined by beta-1,4 glycosidic linkages. Endo-1,4-beta-xylanses are the main enzymes in cleaving beta-1,4 glycosidic bonds from xylan and releasing XOS of varied sizes. XOS, which has a number of important uses in food and in medicine, is largely produced by chemical hydrolysis. More recent research has led to the use of free and immobilized xylanases. This manuscript details the use of a fungal MpXyn10 xylanase recombinantly produced and purified from Aspergillus nidulans and then immobilized on a number of different supports. The researchers found that the MpXyn immobilized on Purolite and on MANAE-agarose retained a significant amount of activity, thermostability, and pH stability when compared to the free enzyme. In addition, the profile of products, typically xylobiose and xylotriose, was consistent across supports, and the XOS yield was ~50% for both the Purolite and MANAE-agarose immobilized enzymes.
Specific Comments:
- A few grammatical and typographical errors need to be resolved.
Thank you. These typographical errors were corrected in the revised manuscript. We also revised the manuscript carefully and corrected any other minor mistakes we found.
- The abstract could be strengthened by indicating the significance of the research detailed in the manuscript. It gets lost why the average reader should be concerned with the production of the XOS by immobilized xylanases.
Thank you for the valuable suggestion. The abstract was improved according to the kind comments of the reviewer. Please see the abstract changed in the revised manuscript.

Reviewer 3 Report
In this work, Immobilization and application of the recombinant xylanase GH10 of Malbranchea pulchella in the production of xylooligosaccharides from hydrothermal liquor of the eucalyptus (Eucalyptus grandis) wood chips is presented. The manuscript is well prepared, however, few queries should be addressed, before considering it for publication:
- As MDPI is pretty straightforward incredibly well-organised journal system, I can't help but wonder, why the authors didn't use the journals' template?
- To continue, there are changes of font style occurring, that need to be removed and corrected throughout the paper, as well as spelling and positions of punctuation marks.
- Activation using epoxy groups, such as with epichlorohydrin should be supported with some additional explanation and literature, such as:
o https://www.sciencedirect.com/science/article/abs/pii/S0141813021004372
o https://www.sciencedirect.com/science/article/abs/pii/S1359511314000543
o https://www.nature.com/articles/s41598-020-76463-x
- Chapter 2.2 is talking about immobilization, so why are there FTIR results presented? They should be presented in chapter 2.3
- It seems that with activity recovery above 100%, such as 142% and 137% for amino-activated, hyperactivation of the enzyme occurs? Should be properly addressed and supported with existing literature, such as:
o https://www.sciencedirect.com/science/article/abs/pii/S1369703X07004111
o https://www.sciencedirect.com/science/article/abs/pii/S0959652618301367
o https://www.sciencedirect.com/science/article/abs/pii/S0141022920302180
- Was characterization performed only with FT-IR? I would suggest to perform at least two additional characterizations using SEM/TEM, XRD, TGA/DSC or DLS to support the occurring improved characteristics of the enzyme after immobilization.
Author Response
Ribeirão Preto, October 24th, 2022.
Dear Dr. Karl Wu
International Journal of Molecular Sciences, Editor
Manuscript Number: IJMS-1972220
We are submitting a revised version of the manuscript entitled " Immobilization and application of the recombinant xylanase GH10 of Malbranchea pulchella in the production of xylooligosaccharides from hydrothermal liquor of the eucalyptus (Eucalyptus grandis) wood chips". We appreciate the attention and the suggestions received to improve our research, and we would like to thank the reviewers for investing their valuable time in reviewing our manuscript. The answers to all questions are listed below. We list the reviewers' comments by numbering them according to their sequence in the reviewers' questions and their respective answers for better understanding. We have revised the entire manuscript and highlighted the changes in red. We have also read carefully through the text and corrected any other minor mistakes we have found.
I look forward to hearing from you.
Dr. Maria de Lourdes T. M. Polizeli
RESPONSES TO REVIEWERS
Reviewer #3:
In this work, Immobilization and application of the recombinant xylanase GH10 of Malbranchea pulchella in the production of xylooligosaccharides from hydrothermal liquor of the eucalyptus (Eucalyptus grandis) wood chips is presented. The manuscript is well prepared, however, few queries should be addressed, before considering it for publication:
- As MDPI is pretty straightforward incredibly well-organised journal system, I can't help but wonder, why the authors didn't use the journals' template?
- To continue, there are changes of font style occurring, that need to be removed and corrected throughout the paper, as well as spelling and positions of punctuation marks.
Thank you. We comply with the reviewer's request. We formatted the revised manuscript according to MDPI guidelines. We also revised the manuscript carefully and corrected any other minor mistakes we have found.
- Activation using epoxy groups, such as with epichlorohydrin should be supported with some additional explanation and literature, such as:
- https://www.sciencedirect.com/science/article/abs/pii/S0141813021004372
- https://www.sciencedirect.com/science/article/abs/pii/S1359511314000543
- https://www.nature.com/articles/s41598-020-76463-x
Thank you for the valuable suggestion. In response to this comment, we addressed the point raised by the reviewer, including the suggested references. Please, see lines 119 to 126 of the revised manuscript.
- Chapter 2.2 is talking about immobilization, so why are there FTIR results presented? They should be presented in chapter 2.3
We comply with the reviewer's request. "Immobilization of the MpXyn10 xylanase" was removed of the Chapter 2.2 and added as new Chapter 2.3. Chapter 2.2 now reads "Preparation and characterization of supports" in the revised manuscript.
- It seems that with activity recovery above 100%, such as 142% and 137% for amino-activated, hyperactivation of the enzyme occurs? Should be properly addressed and supported with existing literature, such as:
- https://www.sciencedirect.com/science/article/abs/pii/S1369703X07004111
- https://www.sciencedirect.com/science/article/abs/pii/S0959652618301367
- https://www.sciencedirect.com/science/article/abs/pii/S0141022920302180
Thank you for the valuable suggestion. In response to this comment, we addressed most of the points raised by the reviewer, including the suggested references. Please, see lines 148 to 182 of the revised manuscript.
- Was characterization performed only with FT-IR? I would suggest to perform at least two additional characterizations using SEM/TEM, XRD, TGA/DSC or DLS to support the occurring improved characteristics of the enzyme after immobilization.
We thank you and agree with the comment. The FT-IR analyzes were carried out to confirm the chemical modification of the supports. The suggestion to improve the characterization with other techniques is very interesting and would certainly add a lot to the manuscript. We understand the raised point; however, the purchase of these data requests’ technical structure and financial and human resources were deeply affected post-pandemic period.

Round 2
Reviewer 1 Report
The authors took into account all my comments. The article can be published in the presented form